# Effect of Hydrothermal Aging on the Tribological Performance of Nitrile Butadiene Rubber Seals

**DOI:** 10.3390/polym16010081

**Published:** 2023-12-26

**Authors:** Gege Huang, Zhihao Chen, Jian Wu, Ange Lin, Qinxiu Liu, Shouyao Liu, Benlong Su, Youshan Wang

**Affiliations:** 1Center for Rubber Composite Materials and Structures, Harbin Institute of Technology, Weihai 264209, China; huanggege@gtc.com.cn (G.H.); 22b908093@stu.hit.edu.cn (Z.C.); 17554179361@163.com (A.L.); 17603881396@163.com (Q.L.); liushouyaoemail@sina.com (S.L.); subenlong@hit.edu.cn (B.S.); wangys@hit.edu.cn (Y.W.); 2Guizhou Tire Co., Ltd., Guiyang 550201, China; 3National Key Laboratory of Science and Technology on Advanced Composites in Special Environments, Harbin Institute of Technology, Harbin 150090, China

**Keywords:** hydrothermal aging, NBR seals, tribological performance, finite element simulation

## Abstract

High temperature and humidity affect the tribological performance of nitrile butadiene rubber (NBR) seals, which affects the precise positioning of cylinder systems. Therefore, it is crucial to study the effect of hydrothermal aging on the tribological performance of the NBR seals. In this study, the changes in the tribological performance of the NBR seals under hydrothermal aging conditions were investigated. The results show that the volatilization of additives and the increase in crosslink density of the NBR seals occurs in the hydrothermal aging environment, leading to the deterioration of their surface quality, elastic deformability, and tribological performance. The formation of surface micropores due to additive volatilization is the main factor in the degradation of tribological performance.

## 1. Introduction

Pneumatic systems are widely used in robotics, aerospace, and medical devices due to their low cost, high efficiency, and light pollution [1,2,3]. The sealing structure is an essential part of realizing the precise positioning control of the cylinder in pneumatic systems, and the variation of the sealing material performance has an important impact on the service performance of the cylinder [4], and in addition, this variation is closely related to the overall efficiency and energy loss of the machine [5,6]. Nitrile butadiene rubber (NBR) is widely used as a sealing material in aerospace, automotive, robotics, etc. due to its high heat resistance, chemical stability, and wear resistance [7].

The rubber materials are subjected to thermal, electrical, optical, and mechanical stresses during service, which leads to rubber aging [8,9,10]. This means that the structure and properties of the rubber have changed, resulting in the deterioration of service performance [11,12,13]. At present, the structure and properties of rubber materials under aging conditions such as thermo-oxidative, prestressed, and marine have been extensively studied [14,15,16,17,18,19]. Moon et al. [20] investigated the aging behavior of rubber materials with a natural rubber/butadiene rubber (NR/BR) system and showed that there are differences in the changes in the structure of natural rubber at different aging stages. At the beginning of the aging process, the aging of rubber materials is dominated by crosslink aging. In contrast, at the later stage of the aging process, its aging mode changes to being dominated by main chain breakage. Hydrogenated nitrile butadiene rubber (HNBR) undergoes similar structural changes during high-temperature aging. The crosslink density on the surface increases after aging, leading to an increase in tensile stiffness and a decrease in failure strain [21]. Liu et al. [22] observed, by swept surface electron microscopy (SEM), that the number of tiny pores appearing on the surface of NBR increased with increasing aging time, accompanied by an increase in the size of the pores. During the aging process, the interior additives of rubber migrate from its interior to the surface [23,24,25]. The research results of Liu et al. [26] on the high-temperature aging of NBR under compression conditions showed that the structural change of NBR during the aging process was mainly cross-linking, while the tensile strength and elongation at the break of nitrile rubber also changed significantly.

The NBR seals in cylinders are subject to complex environmental conditions of service, characterized by high temperature (up to 80 °C) and condensation, resulting in a variety of factors in the process of service by the common influence that inevitably will occur during the hydrothermal aging phenomenon. There are inevitably differences in the tribological performance of the NBR seals due to the hydrothermal aging phenomenon. Researchers have studied the hydrothermal aging of rubber materials to some extent. Choi et al. [27] studied the aging process of sulfur-cured ethylene propylene diene monomer (EPDM) under hydrothermal conditions. They showed that, under hydrothermal conditions, stearic acid in sulfur-cured EPDM reacts with Ca^2+^ in air or water to produce calcium stearate and causes EPDM surface to appear white. Patel et al. [28] investigated the permanent compression deformation of silicone rubber as a function of aging temperature and showed that the silicone rubber aged in a closed system softened with time.

The tribological behavior of rubber materials is a focus of attention [29,30], and the tribological performance after aging is crucial for rubber sealing materials. Dong et al. [31] investigated the dry sliding friction of NBR after aging. The results showed that its wear was dominated by fatigue wear and that aging caused a decrease in tribological performance. Han et al. [32] found that CeO_2_ with a certain amount of graphene could protect the rubber matrix during thermal-oxidative aging and frictional heating. He et al. [33] investigated the tribological performance of CeO_2_ blended rubber after aging and showed that the increase in crosslink density at the beginning of thermal aging could effectively improve its tribological performance. Luo et al. [34] found that the friction and contact pressure of NBR seals increased with the increase in temperature during hydrothermal aging.

However, the effect of hydrothermal aging on the tribological performance of the NBR seals is not fully understood. Therefore, it is particularly important to investigate the effect of hydrothermal aging on the tribological performance of the NBR seals for the practical application of cylinder seal structures. In this work, we investigated the effects of hydrothermal aging times on the mechanical properties and tribological performance of the NBR seals through experimental and finite element simulations to provide theoretical guidance for the subsequent implementation of accurate servo control of cylinders.

## 2. Experimental Methods

### 2.1. Materials

Two types of samples were tested: the NBR cylindrical samples and the NBR seals (PSD-20 from Osaka Co., Ltd., Osaka, Japan). The NBR cylindrical samples were prepared by vulcanizing the raw NBR material. The preparation process is as follows: the raw NBR material was cut into the cylindrical vulcanization mold and placed in the flat plate vulcanizer heated to 160 °C. The vulcanization conditions were 160 °C × 35 min and maintaining the loading pressure of 12 MPa during vulcanization. The specific preparation process is shown in Appendix A. The dimensions of the NBR cylindrical samples were 10 mm in diameter and 10 mm in height, and the NBR seals were tested as a finished part. The specific dimensions of the chosen NBR seals were 20 mm × 14 mm × 2.24 mm (outer diameter × inner diameter × wire diameter).

### 2.2. Hydrothermal Aging Test

The samples were subjected to hydrothermal aging tests. The test method refers to ISO 188-2023 [35]; a TZW-150UVA type Harris environment testing chamber (Wuxi Harris environment equipment Co., Ltd., Wuxi, China) was used for the hydrothermal aging test. The accelerated aging temperature used was 80 °C, the humidity was 85%, and the duration of accelerated aging was 16 days (16 d), which produced the NBR seals with properties similar to those under actual service conditions. The mechanical properties and tribological performance of the samples with different hydrothermal aging times (0 d, 2 d, 4 d, 8 d, and 16 d) were tested.

The NBR seals were tested for tensile permanent deformation under hydrothermal aging conditions, and the test method refers to ISO 2285-2019 [36]. The NBR seals were inserted into cylindrical tensile test devices of different diameters. The tensile rates of the NBR seals were 10%, 20%, 30%, and 40% of the inner diameter, respectively, and the corresponding tensile die diameters were 15.4 mm, 16.8 mm, 18.2 mm, and 19.6 mm. The tensile permanent deformation rates of the NBR seals with different hydrothermal aging times were calculated by Equation (1).
(1)$$K_{1}=\frac{A_{1}-A_{0}}{A_{S}-A_{0}}\times 100\%$$
where *A*_0_ is the original inner diameter length (mm), *A*_1_ is the recovered inner diameter length (mm), and *A_S_* is the tensile inner diameter length (mm). Each sample was measured three times, and the average value was taken.

The NBR cylindrical samples were tested for compression permanent deformation under hydrothermal aging conditions, and the test method refers to ISO 815-1:2019 [37]. The NBR cylindrical samples were compressed by 20% using a compression restrictor. The compressive permanent deformation rates of the NBR cylindrical samples with different hydrothermal aging times were obtained by Equation (2).
(2)$$K_{2}=\frac{h_{0}-h_{2}}{h_{0}-h_{1}}\times 100\%$$
where *h*_0_ is the height of the sample before compression (mm), *h*_1_ is the height of the restrictor (mm), and *h*_2_ is the height of the sample recovery after compression (mm). Each sample was measured three times, and the average value was taken.

### 2.3. Hydrothermal Aged NBR Characterization

The NBR seals with different hydrothermal aging times were intercepted as strips and tensile tests were performed using the uniaxial tensile testing machine (ZQ-990B-5, Zhiqu Precision Instrument Co., Ltd., Dongguan, China). The tests were carried out at a scale distance of 20 mm and a tensile rate of 100 mm/min. The NBR cylindrical samples with different hydrothermal aging times were compressed by the universal testing machine (WDW-02, Sida Testing Technology Co., Ltd., Jinan, China) at a rate of 20 mm/min, and the test method refers to ISO 7743-2017 [38].

The mass of the NBR cylindrical samples and the NBR seals were measured by electronic balance, and each sample was measured four times. The Shore hardness of the NBR cylindrical samples was tested using the LX-A instrument; each sample was measured four times, and the average value was taken as the Shore hardness value of the sample.

The infrared spectra of the NBR cylindrical samples were obtained using a Nicolet 380 Fourier transform infrared spectrometer (FTIR), which was used to compare the functional group composition of the rubber material before and after hydrothermal aging. The spectral range was selected from 800 cm^−1^ to 4000 cm^−1^ in the mid-infrared region with a resolution of 4 cm^−1^, and the number of scans was set to 32. The background was first scanned and then the samples were tested afterward to remove the background. The surface micro-morphology of the NBR cylindrical samples after aging was observed by the optical digital microscope (DSX 510, OLYMPUS Co., Ltd., Japan). The surface of the samples was cleaned with anhydrous ethanol before observation.

### 2.4. Friction Experiment

The tribological tests of the NBR cylindrical samples (0 d–16 d) with different conditions (lubrication and un-lubrication) were tested by the modified friction testing machine shown in Appendix A. The 6061 Aluminum alloy plate was used as the counterpart material, and 612 grease was applied to the surface during the lubrication test. By controlling the normal displacement of the friction testing machine, a load of 5 N was applied to the contact surface of the friction material, and the relative sliding velocity was 20 mm/min. Test the NBR cylindrical samples of 0 d, 1 d, 2 d, 4 d, 8 d, and 16 d, repeat five times for each group of tests, and take the average value.

The cylinder friction experiment machine was used to test the tribological performance of the NBR seals with different hydrothermal aging times and loading air pressures; the test parameters are shown in Table 1.

The cylinder friction experiment machine was modified according to the universal testing machine WDW-02 (Sida Testing Technology Co., Ltd., Jinan, China), and the cylinder type was CDG1BN20-300 (SMC CORPORATION Co., Ltd., Tokyo, Japan). Its structure is shown in Appendix A. The lower chamber of the cylinder was connected to the air pump. The air pressure is provided by the air pump and controlled by the precision regulator (accuracy of 0.001 MPa) to stabilize the loading air pressure. The upper chamber is connected with air to create a pressure difference between the two chambers. The displacement and the change of the external force on the cylinder piston during the test were collected by the displacement sensor and the force sensor at the lower end of the test device, respectively. The test was conducted at room temperature (25 °C ± 2 °C), and the cylinder piston was subjected to uniform reciprocating motion. Each test was performed three times.

The LuGre model is widely used for the description of friction phenomena [39,40,41]. Therefore, the LuGre model was used to describe the dynamic friction of the cylinder, and the state variable A was used to characterize the average deformation of its deflection. The composition of the entire cylinder LuGre model primarily includes the cylinder nonlinear equation of state and the cylinder frictional force equation; the specific form is shown in Equations (3) and (4).
(3)$$\frac{d_{A}}{d_{t}}=v_{0}-k\frac{\left|v_{0}\right|}{g(v_{0})}A$$
(4)$$F_{f}=v_{0}A+\sigma \frac{d_{A}}{d_{t}}+b_{0}v_{0}$$
where *v*_0_ is the piston motion velocity in the cylinder (mm/s); *k* is the axial stiffness of the seal in the cylinder (N/mm); *b*_0_ is the coefficient of viscous friction (N·s/mm^2^); and *σ* is the axial damping coefficient of the seal in the cylinder (N/(mm/s)).

The steady-state form of the LuGre model was mainly determined by the function *g*(*v*_0_); the Stribeck velocity of the cylinder friction is described by Equation (5).
(5)$$g(v_{0})=F_{C}+(F_{S}-F_{C})\exp(-{\left(v_{0}/v_{1}\right)}^{2})$$
where *F_C_* is the dynamic frictional force of the cylinder (N); *F_S_* is the maximum static frictional force of the cylinder (N); and *v*_1_ is the Stribeck speed of the cylinder (mm/s).

The LuGre model assumes that the cylinder is moving at a constant speed, and the deformation of the elastic mane between the two friction contact surfaces reaches a steady state, which means the state variable A is equal to 0. The average deformation value of the model is *g*(*v*_0_)sgn(*v*_0_)/*k* at this time. The cylinder frictional force in the steady state is obtained by Equation (6).
(6)$$F_{w}=\left[F_{C}+(F_{S}-F_{C})\exp(-(v_{0}/v_{1}{)}^{2})\right]\mathrm{sgn}(v_{0})+b_{0}v_{0}$$

According to Equation (6), the cylinder frictional force curve under different piston velocities obtained from the cylinder friction experiment was fitted by the least square method, and the values of four static parameter variables of the LuGre model in the steady state were obtained, which are *F_C_*, *F_S_*, *v*_1_, and viscous friction (*b*_0_*v*_0_). The cylinder friction experiment machine dragged the piston to move at a constant velocity; the frictional force of the cylinder friction experiment can be obtained by Equation (7).
(7)$$F_{f}=F_{1}-(P_{1}A_{1}-P_{2}A_{2})$$
where *F*_1_ is the cylinder piston by all the external force (N), *P*_1_ is the air pressure of the high-pressure chamber (MPa), *P*_2_ is the air pressure of the low-pressure chamber (MPa), *A*_1_ is the area of the piston rod on the high-pressure side (mm^2^), and *A*_2_ is the area of the piston rod on the low-pressure side (mm^2^).

## 3. Finite Element Analysis of the NBR Seals Friction Process

The finite element simulation was conducted using ABAQUS software (Type 6.13). The material parameters were obtained by fitting the results of compression tests. Based on the compressive stress-strain relationship curves of the NBR cylindrical samples at different hydrothermal aging times, the parameters of the Mooney–Rivlin model were obtained, which are shown in Appendix A. The mesh seed was set to 0.1 mm, and the encrypted mesh seed for the contact area at both ends was set to 0.05 mm, using a quadrilateral dominant mesh type. Three analysis steps were set up; the first one was pre-compression of the cylinder wall in contact with the seal, the second one was air pressure through the upper end of the seal, and the third one was the upward movement of the cylinder wall. Restrict the cylinder wall and the inner groove of the cylinder to be rigid bodies. The compression force between the NBR seals and the cylinder wall is reduced due to aging relaxation, and the simulation between the NBR seals and the cylinder wall at different aging times can be achieved by reducing the compression amount of the NBR seals. According to the compression rate of the NBR seals in the cylinder, which is 10% of the inner diameter and the tensile permanent deformation rates of the NBR seals, the equivalent simulation compressions of the NBR seals with different aging times were obtained as 0.050 mm (0 d), 0.042 mm (1 d), 0.034 mm (2 d), 0.031 mm (4 d), 0.026 mm (8 d), and 0.025 mm (16 d).

The simulation model of the NBR seals in the cylinder was established as shown in Appendix A. The cylinder wall and the piston rod were set as analytic rigid with smooth planar, and the NBR seals were set as deformable with Young’s modulus obtained from the compression test of the NBR cylindrical sample and Poisson’s ratio of 0.49. The step type was set as static general. The tangential and normal behaviors are defined in the contact properties. In the tangential behavior, the coefficient of friction (COF) was determined based on the lubrication friction test of the NBR cylindrical samples. In normal behavior, “hard” contact was set. Three analysis steps were set; the first step was the pre-compression of the cylinder wall in contact with the NBR seals, the second step was the air pressure passed to the upper end of the NBR seals, and the third step was the upward movement of the cylinder wall. The specific working parameters for the dynamic friction simulation are shown in Table 2.

## 4. Results and Discussions

### 4.1. Mechanical Properties

Figure 1a,b show the recovered diameters and the tensile permanent deformation rates of the NBR seals with different hydrothermal aging times, respectively. As shown in Figure 1a, the un-tensile NBR seals shrink due to the increase of crosslink and the decrease of molecular chain length in the hydrothermal aging environment. Moreover, the shrinkage of the un-tensile NBR seals is more obvious with the increase of hydrothermal aging time, and the recovered diameter shrank from 20 mm to 19.85 mm when the aging time reached 16 days. The recovered diameters of the NBR seals under different tensile ratios increase with the hydrothermal aging time, which implies that the elastic recovery capacity of the NBR seals under the hydrothermal aging condition is weakened. Moreover, it is found that the difference in recovered diameters between the NBR seals with different tensile rates increased with the hydrothermal aging time, which further proves that the hydrothermal aging condition causes the deterioration of the elastic recovery capacity of the NBR seals.

The tensile permanent deformation rates of the NBR seals in Figure 1b show a monotonically increasing trend with the increase of hydrothermal aging time, which is consistent with the trend in Figure 1a. However, the NBR seals with lower tensile rates have greater tensile permanent deformation rates, which is due to the difference between the recovered diameters of the NBR seals with different tensile rates being smaller than the difference between tensile rates. In addition, it can be found that with the increase of hydrothermal aging time, the increased trend of recovered diameter and permanent deformation rate slows down, indicating that the aging process of NBR slows down with time. Figure 1c shows the variation curve of the compression permanent deformation rates of the NBR cylindrical samples with the hydrothermal aging time. The curve shows an overall trend consistent with that in Figure 1b. When the hydrothermal aging time is 16 d, the compression permanent deformation rate reaches the maximum value of 82.375.

Figure 2a shows the variation curves of the mass of the NBR cylindrical samples and the NBR seals with the hydrothermal aging time. The mass of both samples showed a significant increasing trend, and this trend decreases with the increase of the hydrothermal aging time. The increase in the mass of the NBR under the hydrothermal aging environment is due to the water absorption and moisture absorption reaction. However, small molecular groups, volatile organic compounds, fillers, and additives within the rubber will gradually migrate from the interior to the surface and volatilize under hydrothermal aging conditions [23,24,25], which leads to a small decrease in the mass of the samples. This phenomenon was exhibited in the NBR cylindrical samples of 2 d. Figure 2b shows the variation curves of the hardness of the two samples with the hydrothermal aging time. The hardness of the two rubber material samples exhibits a monotonically increasing trend with the increase of the hydrothermal aging time, and the hardness of the samples exhibits a more obvious increase at the beginning of the hydrothermal aging (0 d–2 d), with an increase of 3.50 HA and 4.58 HA for the rubber cylindrical parts and the NBR seals, respectively. During the hydrothermal aging process, the crosslink density of NBR rubber increased and the internal structure became more compact, which led to an increase in the hardness of the samples.

Figure 3 shows the compression stress-strain curves of the NBR cylindrical samples and the tensile force-deformation curves of the NBR seals with different hydrothermal aging times. As shown in Figure 3, the two samples showed similar trends in the compression test and tensile test; the NBR cylindrical samples and the NBR seals exhibited enhanced compressive strength and tensile strength, respectively, with the hydrothermal aging time increased. This enhanced trend became more obvious with the increase of hydrothermal aging time. The increase in the compressive strength and tensile strength of the samples is due to the increase in the crosslink density of NBR in the hydrothermal aging environment [21,26]. The degree of connection between the molecular network chains is enhanced, and more molecular chains bear stress during compression or stretching. In addition, the increase in the degree of crosslink between molecules leads to the shortening of the molecular chain and the deformation amount of the NBR decreases when subjected to stress. The other test data are shown in Appendix A.

### 4.2. Tribological Behavior of Hydrothermal Aging

Figure 4 shows the surface morphology of NBR cylindrical samples with different hydrothermal aging times. The surface quality of the NBR cylindrical samples deteriorated due to hydrothermal aging, and obvious microporous structures appeared. Moreover, the number and size of these micropores increase significantly with the increase of hydrothermal aging time, which means that the deterioration of its surface quality is positively correlated with the hydrothermal aging time. This deterioration is mainly due to the migration of fillers, small molecular groups, and volatile organic compounds, etc., in the rubber material from internal to surface and volatilization [23,24,25], which is consistent with the results in Figure 2 and Figure 4. The deterioration of the frictional surface quality is an important factor leading to the reduction of the tribological performance of the rubber material [42].

During the service of cylinder seals, the loading air pressure in the cylinder is an important environmental condition; therefore, the tribological performance of cylinder seals under different loading air pressure is studied first. The cylinder friction experiment was conducted to further investigate the actual effect of hydrothermal aging on the tribological performance of the NBR seals in the cylinder. The cylinder friction experiment needs to be tested from the highest velocity and then reduced to the experimental velocity. In the low-velocity cylinder friction experiment, the cylinder needs to be stared at high velocity three times, which ensures the internal grease distribution is uniform.

The influence of the 0 d NBR seals on the static parameters of the cylinder was investigated. As shown in Figure 5a, the *F_f_*-piston velocity variation curves and fitted curves are shown for the cylinder equipped with the 0 d NBR seals. Based on Equation (6), the squared difference between the curve obtained by least squares fitting and the mean value of the experimental results and the fitting coefficient is 0.9974, which has a good fitting effect. According to Figure 5b, which shows the effect of the loading air pressure on the *F_C_* of the cylinder equipped with the 0 d NBR seals, it can be seen that with the increase of the loading air pressure, the static parameters in the friction model changed accordingly, and the *F_C_* showed a monotonically increasing trend. When the loading air pressure increases from 0 MPa to 0.35 Ma, the *F_C_* increases from 6.7 N to 10.5 N (with a total increase of about 56.72%). This indicates a linear relationship between increasing loading air pressure and the dynamic tribological performance of the cylinder. After investigating the kinetic tribological performance of the cylinder with different loading air pressure and the tribological performance of the 0 d NBR seals, the kinetic tribological performance of the cylinder with the aged NBR seal was further investigated.

The friction tests were conducted on cylinders equipped with the aged NBR seals at different velocities in 0 MPa, and the results are shown in Appendix A. The curves were fitted by the least squares method, using Equation (6), in order to obtain the variation curve of *F_C_* with the hydrothermal aging times, as shown in Figure 6a. It can be observed that the *F_C_* increases not linearly with the increase of hydrothermal aging time at the loading air pressure of 0 MPa. According to the extension line, the incremental rate of *F_C_* increases gradually with the increase of hydrothermal aging time, which indicates that the relationship between hydrothermal aging time and *F_C_* shows an exponential change, and the increase of hydrothermal aging time leads to a faster decrease of tribological performance of the NBR seals. As shown in Figure 6b, the *F_C_* increases with the loading air pressure, which does not change due to hydrothermal aging. Moreover, the variation trend of *F_C_* with hydrothermal aging time does not change with the loading air pressure. This indicates that the loading air pressure does not affect the effect of hydrothermal aging on the tribological performance of the NBR seals in the cylinder during service, which only generates greater friction during the cylinder motion.

The mechanical test proved that the mechanical properties of the NBR cylindrical samples and NBR seals vary in a consistent manner, and their compositions are both NBR. In order to further investigate the influence of loading air pressure on the tribological performance of the NBR seals, the frictional force and the contact area of the NBR seals under different simulation conditions were first analyzed by finite element simulation.

Figure 7 shows the frictional force and the contact area of 0 d NBR seals. Figure 7a shows the frictional force of the 0 d NBR seals under different loading air pressures. The frictional force shows a monotonic increasing trend with the increase of loading air pressure and is consistent with the trend of *F_C_* variation in the cylinder friction experiment. The frictional force of the NBR seals increases by 1.54 N when the loading pressure increases from 0 MPa to 0.35 MPa. Figure 7b shows the contact areas between the 0 d NBR seals and the cylinder wall under different loading air pressures. The contact areas increase with the loading air pressure, and this increasing trend is phased. The contact areas increase rapidly at the loading air pressure of 0.05 MPa, 0.2 MPa, and 0.3 MPa, the loading air pressure increased from 0 MPa to 0.35 Ma, and the contact area increased from 22.1 mm^2^ to 31.4 mm^2^, which is a total increase of about 42.1%. This indicates that the variation of the frictional force of the NBR seals with the loading air pressure is not completely determined by the contact area. In order to further investigate the variation of the tribological performance, the variation of the stress at different loading air pressures was analyzed.

Figure 8 shows the Mises stress distribution cloud and shear stress (CSHEAR1) distribution at the frictional contact interface of the 0 d NBR seals. The maximum Mises stress of the seal gradually shifts to the lower right side of the NBR seal with the increase of the loading air pressure and gradually increases. Meanwhile, the NBR seals move downward and deform significantly as the loading air pressure increases and their contact position with the cylinder wall shifts from the middle region to the lower region. The shear stress of the NBR seals presents a parabolic shape and reaches the maximum value at the center of the frictional contact interface. The change of shear stress of the NBR seals increases significantly with the increase of the loading air pressure. The maximum shear stress increases from 0.037 MPa to 0.084 MPa when the air pressure increases from 0 MPa to 0.35 MPa. This is mainly due to the increased loading air pressure causing the NBR seals to have a greater tendency to deform and therefore generate greater contact pressure at the frictional contact interface, which can be seen in the Mises stress distribution cloud.

Meanwhile, according to the distribution of shear stress, it is found that the value is smaller at the edge of the contact interface, and the frictional force does not show a phased increase when the contact area phase increases, which indicates the change of contact area does not determine the frictional force of the NBR seals under different loading air pressure. Moreover, the change of maximum shear stress shows the same trend as the change of frictional force, and it can be judged that the effect of loading air pressure on the frictional force of the NBR seals in the cylinder is mainly through the increase of contact pressure at the contact interface, not the increase of contact area. Appendix A show the shear stress distribution of the NBR seals under different hydrothermal aging times. Hydrothermal aging has no obvious effect on the distribution pattern of shear stress at the friction contact interface of the NBR seals. The effect of hydrothermal aging on the shear stress is mainly reflected in the distribution area and size of the shear stress. This change is mainly due to the hydrothermal aging caused by the elastic modulus of the NBR seals increasing, thus reducing the friction process of the contact area, and is caused by the increase in shear stress.

Figure 9 shows the variation of frictional force and contact area of the NBR seals with different loading air pressures for different hydrothermal aging times. According to Figure 9a, the variation trend of aging NBR seals obtained by finite element simulation at different loading air pressures is generally consistent with the *F_C_* results of the cylinder friction experiment. However, there are some differences, which will be analyzed later. According to Figure 9b, the contact area of the aged NBR seals during the friction process exhibits a phased increase with the increase of loading air pressure (specific values are shown in Appendix A), which is consistent with the trend of the 0 d NBR seals. This indicates that hydrothermal aging does not change the influence of loading air pressure on the tribological performance of the NBR seals, which is consistent with that of the cylinder friction experiment. The frictional force of the NBR seals obtained in the finite element simulation showed a certain degree of decrease within 0 d–2 d. This decreasing trend disappeared with the increase of the loading air pressure, and when the loading air pressure increased to 0.3 MPa, the trend of its frictional force was basically the same as that of the *F_C_* in the cylinder friction experiment. The NBR seals inevitably shrink due to the increase of cross-link density in the hydrothermal aging environment; its compression in the cylinder decreases and leads to the decrease of contact pressure, which decreases the frictional force in 0 d–2 d. However, the COF of the NBR seals increases with the hydrothermal aging time, and, after 2 d, the frictional force shows a trend of increasing with the hydrothermal aging time. Meanwhile, the frictional contact area of the NBR seals increases with the loading air pressure increases, which implies that the contact pressure increases, and effectively reduces the reduction of the contact pressure between the NBR seals and the cylinder wall due to hydrothermal aging. Therefore, the decreasing trend of the frictional force decreases with the increase of loading air pressure within 0 d–2 d. This can be further confirmed by comparing the multiplication of the increase in COF in Appendix A and the increase in frictional force in Figure 9 with hydrothermal aging time, revealing that the multiplication of the increase in COF is greater than the increase in frictional force. The shrinkage of the inner diameter of the NBR seals caused by hydrothermal aging reduces its contact pressure with the cylinder wall and suppress the decrease of its tribological performance, which is known by combining the above analysis and Figure 1.

In addition, it was found that the main difference between the cylinder friction experiment and the finite element simulations is the surface quality of the NBR seals. According to the friction test of the NBR cylindrical samples and previous studies [42], the surface quality of the rubber material is closely related to its tribological properties. Moreover, the frictional contact area is not the main factor affecting the tribological performance according to the cylinder friction experiment and the finite element simulation results, and the shrinkage of NBR seals in the hydrothermal aging environment reduces the contact pressure with the cylinder wall. Therefore, the effect of damp aging on the surface quality of NBR seals leads to the main factor of its tribological performance change. The evaporation of additives inside the NBR in the hydrothermal aging environment causes the deterioration of the surface quality and leads to a faster deterioration of the tribological performance with increasing hydrothermal aging time.

## 5. Conclusions

The influence law of hydrothermal aging conditions on the mechanical properties and tribological performance of rubber sealing materials was investigated through experiments and finite element simulation. This has an important theoretical value and significance for the subsequent realization of servo control of pneumatic servo systems under hydrothermal aging conditions and the development of pneumatic technology. The main conclusions are summarized as follows:NBR undergoes oxidation, cross-linking, chain-breaking, etc. in the hydrothermal aging environment, which leads to changes in the chemical structure. Moreover, the internal additives precipitate out onto its surface, which causes the appearance of surface micropores. The number and size of these micropores increase with the hydrothermal aging time.The cross-link density between molecules of NBR increases in the hydrothermal aging environment, and the strength of the connection between molecular chains is enhanced, resulting in increased hardness, tensile and compressive resistance, and weakened elastic deformation. Therefore, the NBR seals shrink in the hydrothermal aging environment, which reduces the contact pressure between them and the cylinder wall.The increase in loading air pressure in the cylinder causes an increase in dynamic friction and does not affect the effect of hydrothermal aging time on the tribological performance of NBR seals. The dynamic friction of the NBR seals in the cylinder gradually increases with the increase of the hydrothermal aging time, which is mainly due to the increase in the number and size of surface micropores caused by the volatilization of additives inside the NBR in the damp aging environment. The deterioration of the surface quality leads to the degradation of the tribological performance of the NBR seals, and shrinkage of the NBR seals due to hydrothermal aging suppresses the degradation of the tribological performance.

## Figures and Tables

**Figure 1 polymers-16-00081-f001:**
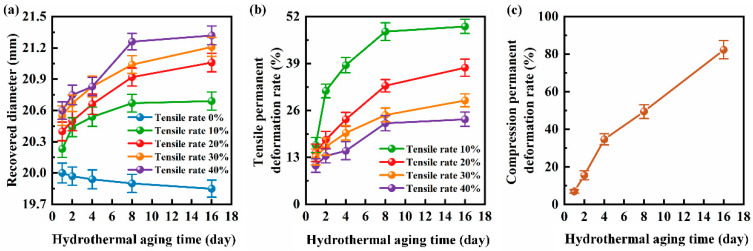
(**a**) The recovered diameters, (**b**) tensile permanent deformation rates of the NBR seals, and (**c**) the compression permanent deformation rates of the NBR cylindrical samples with different hydrothermal aging times.

**Figure 2 polymers-16-00081-f002:**
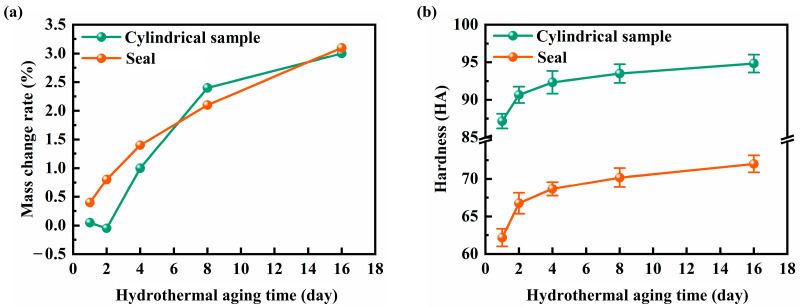
The variation curves of (**a**) mass and (**b**) hardness of the NBR cylindrical samples and the NBR seals with different hydrothermal aging times.

**Figure 3 polymers-16-00081-f003:**
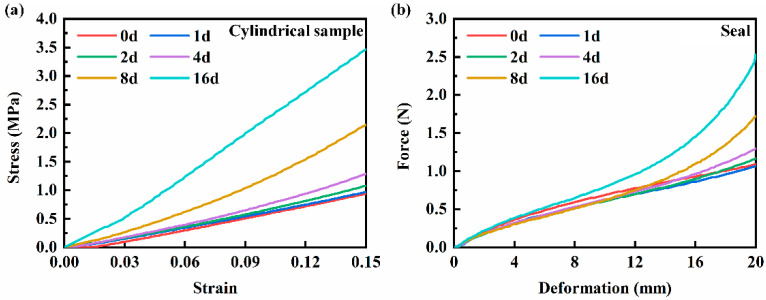
(**a**) Stress-strain curves of the NBR cylindrical samples and (**b**) force-deformation curves of the NBR seals with different hydrothermal aging times.

**Figure 4 polymers-16-00081-f004:**
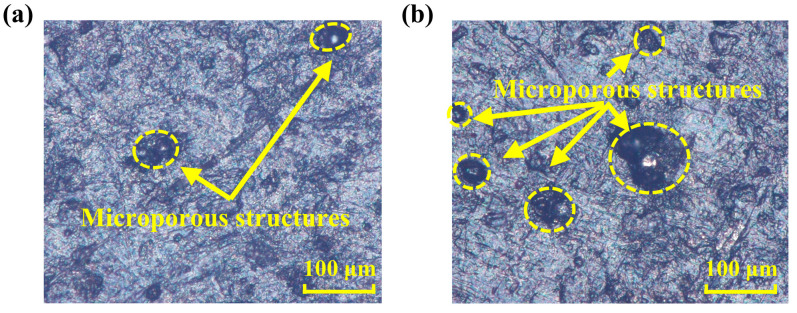
The surface morphology of the (**a**) 1 d and (**b**) 16 d NBR cylindrical samples.

**Figure 5 polymers-16-00081-f005:**
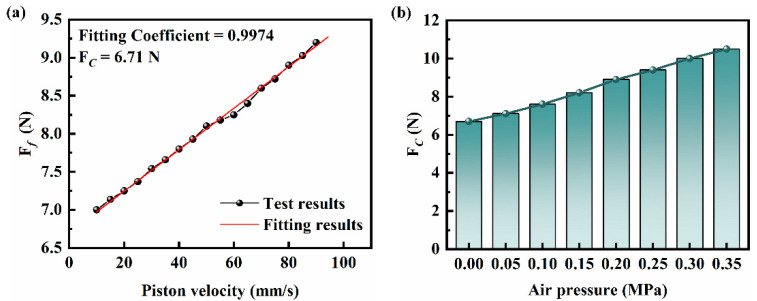
(**a**) The *F_f_*-piston velocity variation curves and fitted curves of the cylinder equipped with the 0 d NBR seals; (**b**) the *F_C_* of the cylinder equipped with the 0 d NBR seals with different loading air pressure.

**Figure 6 polymers-16-00081-f006:**
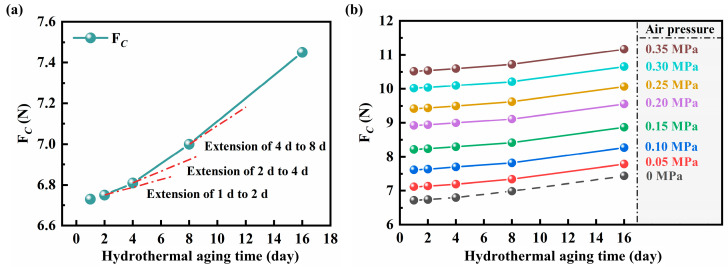
The *F_C_* of the cylinder equipped with the aged NBR seals under loading air pressure of (**a**) 0 MPa and (**b**) 0–0.35 MPa.

**Figure 7 polymers-16-00081-f007:**
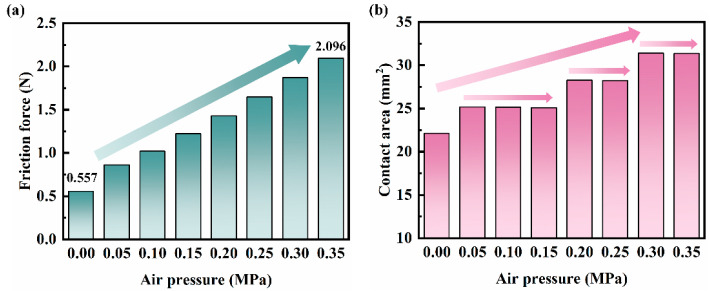
(**a**) The frictional force and (**b**) the contact area of the 0 d NBR seals under different loading air pressures.

**Figure 8 polymers-16-00081-f008:**
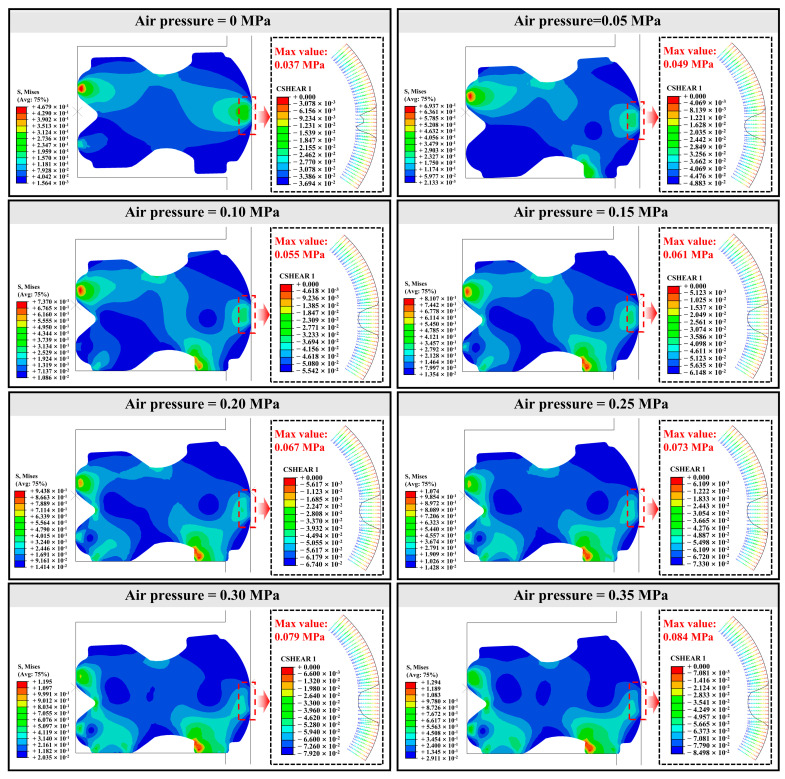
The Mises stress clouds and shear stress (CSHEAR1) distribution of the 0 d NBR seals under different loading air pressures.

**Figure 9 polymers-16-00081-f009:**
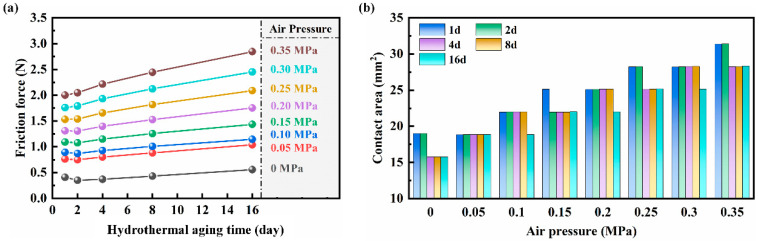
(**a**) The frictional force and (**b**) the contact area of the aged NBR seals under different loading air pressures.

**Table 1 polymers-16-00081-t001:** The test parameters of the cylinder friction experiment.

Aging Time (Day)	Loading Air Pressure (MPa)	Piston Velocity (mm/s)
0	0, 0.05, 0.1, 0.15, 0.2, 0.25, 0.3, 0.35	10, 15, 20, 25, 30, 35, 40, 45, 50, 55, 60, 65, 70, 75, 80, 85, 90
1		
2		
4		
8		
16		

**Table 2 polymers-16-00081-t002:** Finite element simulation conditions.

Aging Time (Day)	Loading Air Pressure (MPa)
0	0, 0.05, 0.1, 0.15, 0.2, 0.25, 0.3, 0.35
1	
2	
4	
8	
16	

## Data Availability

Data are contained within the article and Appendix A.

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
