# Peer review of "Effect of Hydrothermal Aging on the Tribological Performance of Nitrile Butadiene Rubber Seals"

_polymers, 2023, doi:10.3390/polym16010081_

Round 1

Reviewer 1 Report

Comments and Suggestions for Authors

The authors show a very interesting work on this hydrothermal aging topic in combination with tribological aspects. However, some improvements of the manuscript should be done:

In the introduction sometimes hydrothermal and hygrothermal ageing is meant. Please clarify if it is no typo! 

Line 144: For tribological testing it is important to clarify the sour face roughness of the counterpart and test specimen 

Line 145: Could you please give more information about 612 grease? 

Line 197: Information about the mesh size and selected type is missing. In general the model with ist boundary conditions is not very well described in the supplementary material. Please improve! 

Line 213: Where did you get the Poisson`s ratio value for NBR?

Figure 1. It seems that all standard-deviations look similar. Could that be? Please check! Additionally from the small figure size they are also not clear readable.

Line 226: How was the hardness measured? 

Figure 3: Please show at least in the supplementary section the second and third test of each aging state.

Figure 4: When comparing two different spectra’s it should be normed to a specific band. From this view it is not clear if the characteristic bands are really decreasing 

Line 324: In the whole section values regarding the COF and also the standard-deviation is missing! This is also important due to that fact that these values are used for simulation.  

Figure 10b: The increase of contact area due to increase of pressure is clear but the there is no clear trend regarding the aging times. Could please comment on that? Also the interpretation of the contact area is no covering surface roughness. 

Reviewer 2 Report

Comments and Suggestions for Authors

The paper of Wu et al. “Effect of Hydrothermal Aging on the Tribological Performance 2 of Nitrile Butadiene Rubber Seals” deal with the study of tribological performance after hydrothermal aging.

The study is interesting and can be published after minor revision and simple adding investigations.

I recommended to measure the hardness of neat samples and its after hydrothermal aging because this parameter is more important for tribology than elongation or tensile strength.

I recommended in more details to describe a FTIR experiments. For me is clear that authors use the ATR method for recording of IR spectra. Therefore, the hardness of sample surface is important for recording of IR spectrum. If look on IR spectra in paper (Figure 4) can see that IR spectrum of sample before hydrothermal aging is better resolved than spectrum of sample after. That made speculative all suggestions about changing of chemical structure. By my opinion the means about formation of hydroxyl groups on the surface (line 291) is mistake because as one can band of 3357 cm-1 present in neat sample too and it origin connect with absorbed water.

I recommended to delete all presentation about IR study because all in this presentation is to speculative and did not contain important information for tribological study.    

Comments on the Quality of English Language

non

Reviewer 3 Report

Comments and Suggestions for Authors

The manuscript is interesting. However some minor issues are found. 

The synthesis method is not fully clear.

Explain the role of Hydrothermal aging test.

Is there any effect of Hydrothermal aging test on surface morphology? Explain.

Is shear stress distribution connected to hydrothermal aging?

What is the novelty here?

Mentioned the motivation behind this work.

Comments on the Quality of English Language

minor
